# Safety and efficacy of intravenous or topical tranexamic acid administration in surgery: a protocol for a systematic review and network meta-analysis

Xinyan Wang, Xinxin Wang, Fa Liang, Yun Yu, Ruquan Han 

Department of Anesthesiology, Beijing Tiantan Hospital, Capital Medical University, Beijing, China

**Correspondence to**
Dr Ruquan Han;
ruquan.han@gmail.com

## ABSTRACT

**Introduction** Tranexamic acid (TXA) has become a widely used antifibrinolytic drug for reducing bleeding in surgery. However, adverse events, such as seizures, pulmonary embolism and deep vein thrombosis, limit its application. To date, insufficient attention has been devoted to determining the optimal dosage and administration route of TXA in the field of surgery. Thus, this study uses the network meta-analysis method, relying on its characteristics of combining direct comparison and indirect comparison, to analyse the safety and efficacy of different doses (high, medium, low) of intravenous injection or of topical application of TXA.

**Methods and analysis** We will search the PubMed, Cochrane Central Register of Controlled Trials, Embase, Web of Science and China National Knowledge Internet databases using a strategy that combines the terms TXA, randomised controlled trials and embolism (or haemorrhage, blood transfusion, seizure, mortality). Two reviewers will independently screen all identified abstracts for eligibility and evaluate the risk-of-bias of the included studies using the Cochrane risk of bias tool for randomised controlled studies. We will conduct a systematic review and network meta-analysis. We plan to investigate heterogeneity by performing subgroup analysis and sensitivity analysis, and we will also consider the dose–response relationship between the optimal dose and a better routine. We will assess the overall certainty of the evidence for each outcome using the Grading Recommendations Assessment, Development and Evaluation approach

**Ethics and dissemination** No ethics approval will be sought, as no original data will be collected for this review. Findings will be disseminated through peer-reviewed publications and conference presentations.

**PROSPERO registration number** CRD42021281206.

| Strengths and limitations of this study |
| --- |
| ⇒ This is the first network meta-analysis to focus on the optimal dose and administration route of tranexamic acid (TXA) in surgery. |
| ⇒ Unpublished ongoing clinical studies will be searched using the WHO International Clinical Trials Registry Platform and ClinicalTrials.gov. |
| ⇒ The quality of evidence will be assessed using the Grading Recommendations Assessment, Development, and Evaluation. |
| ⇒ We will have no language restrictions in this meta-analysis, thus yielding a more comprehensive group of eligible studies. |
| ⇒ The main limitation of our study protocol is that some of the included trials may not be of high quality, which will influence our assessment of the safety and efficiency of the types of TXA administration. |

application of personalised, evidence-based, care bundles of interventions that reduce bleeding and transfusion to improve clinical outcomes,[6–8] such as autologous blood transfusion, restricted blood transfusion, point-of-care diagnostic test-based algorithms for the personalised treatment of coagulopathy and antifibrinolytic drugs.[6–8] Among them, the antifibrinolytic drug tranexamic acid (TXA) has become widely used.

In 2021, two systematic reviews and meta-analyses of TXA were published in *Annal of Surgery*.[2 9] The results showed that perioperative prophylactic topical use or single-dose intravenous TXA has high safety and a low incidence of adverse events.[2 9] Nonetheless, there are still adverse events, such as seizures, pulmonary embolism, deep vein thrombosis, headache and fatigue.[10 11] Previous research has established that postoperative seizures are associated with TXA dosage,[12] which may be because of the continuous high concentration of TXA in the brain tissue at the early stage after surgery.[13] Thromboembolic events are another major concern,[11 14] especially in

## BACKGROUND

Bleeding is a major problem in surgery.[1 2] However, the common treatment—transfusion—is limited by a lack of blood sources, high processing costs, adverse blood transfusion reactions and transmission of infectious diseases.[3–5] Recently, the concept of patient blood management has attracted more attention.[6 7] It has been defined as the timely

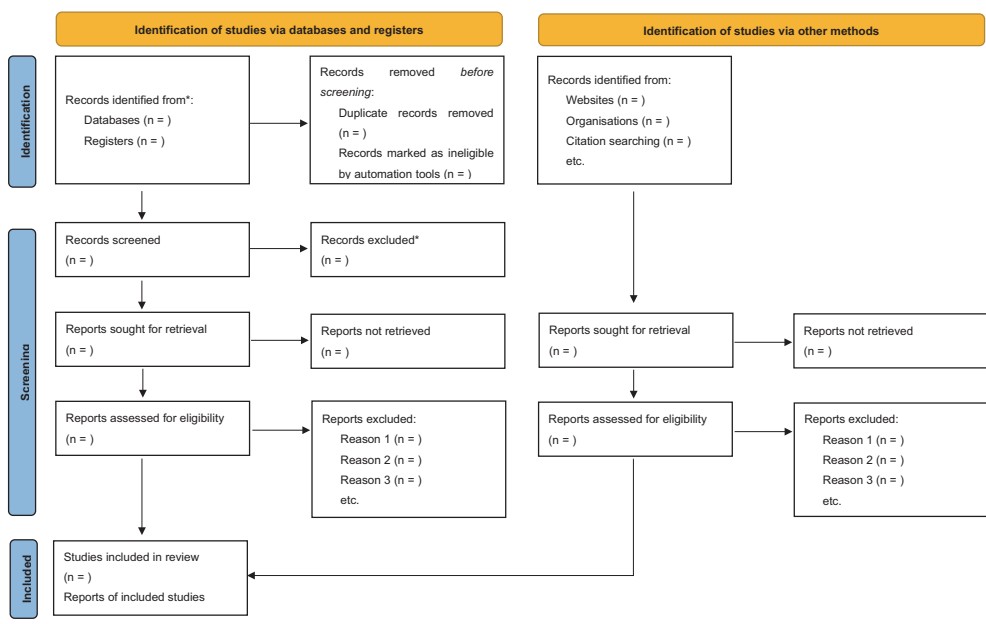

**Figure 1** Flow chart diagram presents the selection of articles for systemic review and network meta-analysis of the safety and efficacy of intravenous or topical tranexamic acid administration in surgery. *Consider, if feasible to do so, reporting the number of records identified from each database or register searched (rather than the total number across all databases/registers).

patients who are in a hypercoagulable state due to stress during the perioperative period.[15 16] TXA, as an antifibrinolytic drug, tends to increase coagulation function.[17] Therefore, adopting the appropriate route of administration and dosage has become the primary solution strategy.

The most popular clinical administration routes of TXA are intravenous infusion and topical use.[9 17 18] To date, far too little attention has been given to the optimal dosage and administration route of TXA in the field of surgery.[19] Conducting multidose, multiple intervention, large-scale, randomised controlled studies is time-consuming and labour intensive. In this network meta-analysis, we plan to analyse the safety and efficacy of different doses (high, medium, low) of intravenous injection or topical application of TXA in surgical patients of all ages using direct comparison and indirect comparison.[20]

## METHODS AND ANALYSES

The review has been registered in PROSPERO International Prospective Register of Systematic reviews (https://www.crd.york.ac.uk/PROSPERO) and will be reported in accordance with the Preferred Reporting Items for Systematic reviews and Meta-Analysis protocol (PRISMA-P) guidelines.[21] The PRISMA flow diagram will be used to record every step of the review process (figure 1). We are planning to start the review in September 2021 and to complete it in December 2022 at the latest.

## SOURCES OF EVIDENCE AND SEARCH STRATEGY

We will include all published studies fulfilled with criteria without language restriction in this protocol. The published studies will be identified by searching the PubMed, Cochrane Central Register of Controlled Trials, Embase, Web of Science and China National Knowledge Internet databases from inception to 20 September 2021 (tables 1–5). Unpublished ongoing clinical studies will be identified by searching the WHO International Clinical Trials Registry Platform and ClinicalTrials.gov up to 20 September 2021. We will continue to update and include the latest articles that meet the search criteria until the deadline. The two researchers (XinyW and XinxW) will independently conduct the literature searches in accordance with the search strategy ("Tranexamic acid", "Randomized Controlled Trial" and one of the outcomes, specific strategy in tables 1–5).

### Inclusion and exclusion criteria

To be included in this systematic review, studies must fulfil each of the criteria outlined below.

### Research type (S)

Randomised controlled trial.

**Table 1** Search strategy of PubMed

| Search number | Query |
| --- | --- |
| #1 | Tranexamic Acid [MeSH Terms) |
| #2 | (((((((((((AMCHA(Title/Abstract)) OR (trans-4-(Aminomethyl)cyclohexanecarboxylic Acid(Title/Abstract))) OR (t-AMCHA(Title/Abstract))) OR (AMCA(Title/Abstract))) OR (Anvitoff(Title/Abstract))) OR (Cyklokapron(Title/Abstract))) OR (Ugurol(Title/Abstract))) OR (KABI 2161(Title/Abstract))) OR (Spotof(Title/Abstract))) OR (Transamin(Title/Abstract))) OR (Amchafibrin(Title/Abstract))) OR (Exacyl(Title/Abstract)) |
| #3 | #1 or #2 |
| #4 | Randomized Controlled Trials as Topic [MeSH Terms] |
| #5 | ((((Clinical Trials, Randomized(Title/Abstract)) OR (Trials, Randomized Clinical(Title/Abstract))) OR (Controlled Clinical Trials, Randomized(Title/Abstract))) OR (Clinical trials(Title/Abstract))) OR (Randomized Controlled*(Title/Abstract)) |
| #6 | #4 or #5 |
| #7 | "Embolism"(MeSH Terms] OR "Embolisms"(Title/Abstract)OR "Embolus"(Title/Abstract)OR "thromboembolism"(MeSH Terms] OR "Thrombosis"(Mesh] OR "Venous Thrombosis"(Mesh) |
| #8 | "hemorrhage"(MeSH Terms] OR "Hemorrhages"(Title/Abstract)OR "Bleeding"(Title/Abstract) |
| #9 | "Blood transfusion"(MeSH Terms] OR "blood transfusion"(MeSH Terms] OR "blood transfusion"(MeSH Terms) |
| #10 | "Seizures"(MeSH Terms] OR "Seizure"(Title/Abstract)OR "Mortality"(MeSH Terms] OR "Mortalities"(Title/Abstract) OR "death rate"(Title/Abstract)OR "case fatality rate"(Title/Abstract)OR "crude mortality rate"(Title/Abstract) |
| #11 | #7 or #8 or #9 or #10 |
| #12 | #3 and #6 and #11 |

## Participant (P)

### Inclusion criteria

1. Patients (regardless of age) undergoing surgery, including orthopaedics, neurosurgery, obstetrics and gynaecology, plastic surgery, paediatrics.
2. Intravenous or topical use of TXA, indicating the dosage.
3. Including at least one of the following outcome indicators: blood loss (intraoperative, postoperative or total blood loss); blood transfusion; blood transfusion rate; thromboembolic events (deep vein thrombosis, pulmonary embolism), seizures, death.

### Exclusion criteria

Studies that did not meet the inclusion criteria, such as:

1. Inconsistent research types: cohort studies, case–control studies, case reports, reviews.
2. Lack of outcome indicators, lack of OR and standardised mean difference data

### Intervention and groups

1. Low topical TXA, ≤1 g.
2. Medium doses of topical TXA, 1–2 g.
3. High doses of topical TXA, >2 g.
4. Intravenous low-dose TXA was defined as an infusion dose≤1 g, an initial dose≤10 mg/kg or a maintenance dose≤10 mg/kg/h.
5. Intravenous medium-dose TXA: infusion dose 1–2 g, or initial dose 10–20 mg/kg, maintenance dose≤15 mg/kg/h.
6. Intravenous high-dose TXA: infusion dose≥2 g, initial dose>20 mg/kg, maintenance dose≤20 mg/kg/h.
7. Placebo.

## Outcomes

1. Efficacy outcomes: bleeding volume, blood transfusion volume and blood transfusion rate, operation duration, postoperative Hb and Hct values, change in Hb or Hct, and drain output.
2. Safety outcomes: mortality, incidence of thromboembolism, seizures and haematoma.

## Languages

There will be no language restriction. We will seek professional translators with bilingual language backgrounds for translation. For example, when we encounter an article in French, we will find a professional translator proficient in French and Chinese (or French and English) to translate the original text to ensure the accuracy of the translation.

## Time

The anticipated start date is September 2021, and the anticipated completion date is December 2022.

## Study records

### Data management

The results of the literature search will be imported into the EndNote X V.9.3.3 database, and duplicates will be removed. We will establish several independent groups for each selecting stage in the EndNote database. Abstracts and full-text articles will be uploaded to the database. The extraction information table of the final included studies has been designed, and the study team will receive training.

### Selection process

We (XinyW and XinxW) will independently screen studies according to the inclusion and exclusion criteria. The

**Table 2**  Search strategy of the Cochrane Central Register of Controlled Trials

| Search number | Query |
|---|---|
| #1 | MeSH descriptor: [Tranexamic Acid] explode all trees |
| #2 | (AMCHA or t-AMCHA or AMCA or trans-4 Aminomethyl-cyclohexane carboxylic Acid or KABI 2161 or Cyklokapron or Transamin or Spotof or Ugurol or Exacyl or Anvitoff or Amchafibrin): ti, ab, kw (Word variations have been searched) |
| #3 | #1 or #2 |
| #4 | MeSH descriptor: (Randomised Controlled Trial) explode all trees |
| #5 | (Randomized Control* or clinical trial*): ti, ab, kw (Word variations have been searched) |
| #6 | #4 or #5 |
| #7 | MeSH descriptor: [Haemorrhage] explode all trees |
| #8 | (Bleeding or Hemorrhages): ti, ab, kw (Word variations have been searched) |
| #9 | #7 or #8 |
| #10 | MeSH descriptor: [Blood Transfusion] explode all trees |
| #11 | (“transfusion”): ti, ab, kw (Word variations have been searched) |
| #12 | #10 or #11 |
| #13 | MeSH descriptor: [Mortality] explode all trees |
| #14 | (Mortality Decline or Death Rate or Crude Mortality Rate): ti, ab, kw (Word variations have been searched) |
| #15 | #13 or #14 |
| #16 | MeSH descriptor: [Thromboembolism] explode all trees |
| #17 | MeSH descriptor: [Embolism] explode all trees |
| #18 | (Embolisms or Embolus): ti, ab, kw (Word variations have been searched) |
| #19 | MeSH descriptor: [Thrombosis] explode all trees |
| #20 | MeSH descriptor: [Venous Thrombosis] explode all trees |
| #21 | MeSH descriptor: [Seizures] explode all trees |
| #22 | #16 or #17 or #18 or #19 or #20 or #21 |
| #23 | #9 or #12 or #15 or #22 |
| #24 | #3 and #6 and #23 |

merged results will be imported into Endnote X V.9.3.3, and duplicate studies will be removed.

All searched articles will be selected in a two-stage process. First, the title and abstract will be assessed based on the inclusion and exclusion criteria. The study will be removed if it does not meet the criteria. Next, full texts of articles retained in the first round of screening will be retrieved and examined based on eligibility criteria to confirm their inclusion, and studies that do not fulfil the criteria will be removed.

Both steps of the assessment will be performed independently by two reviewers (XinyW and XinxW). If an inconsistency occurs, a third reviewer will be consulted. We will record reasons for exclusion at both stages of the inclusion process.

## Data extraction
We (XinyW and XinxW) will independently extract the following data from each study: study type, participants, inclusion criteria, exclusion criteria, baseline characteristics (age, sex, etc), country, setting, interventions, all outcomes, findings and study dates. After data extraction, we will compile the information and import it into Excel spreadsheets. When an inconsistency occurs, we will recheck the original document to correct the error. When dealing with missing data, we will contact principal investigators to obtain unreported data or other detailed information.

## Risk-of-bias assessment
We (XinyW and XinxW) will evaluate the risk of bias for each included study independently. The Cochrane risk-of-bias tool is used to assess randomised controlled studies.[22] The evaluation scale will import into Revman software in advance, and specific reasons will be provided for each evaluation characteristic. If an inconsistency occurs, a third reviewer (Fa Liang, Yun Yu or Ruquan Han) will be consulted.

## Statistical analysis
The network meta-analysis will be performed using STATA V.13.1, Revman V.5.3, and R software V.3.6.0. Risk ratios with 95%CIs will be calculated using the random effects model for investigating treatment effects. A z test will be conducted to assess the significance of the overall effect size. A p value of <0.05 will be considered statistically significant.

**Table 3** Search strategy of Embase

| Search number | Query |
|---|---|
| #1 | 'Tranexamic acid'/exp |
| #2 | ((('amcha'/exp OR amcha OR 't amcha' OR 'amca'/exp OR amca OR 'trans 4') AND 'aminomethyl cyclohexane' AND carboxylic AND ('acid'/exp OR acid) OR kabi) AND 2161 OR 'cyklokapron'/exp OR cyklokapron OR 'transamin'/exp OR transamin OR spotof OR 'ugurol'/exp OR ugurol OR 'exacyl'/exp OR exacyl OR 'anvitoff'/exp OR anvitoff OR 'amchafibrin'/exp OR amchafibrin) AND (abstracts)/lim |
| #3 | #1 OR #2 |
| #4 | 'Randomized controlled trial (topic)'/exp |
| #5 | 'Clinical trial (topic)' |
| #6 | #4 OR #5 |
| #7 | hemorrhage |
| #8 | 'bleeding'/exp |
| #9 | #7 OR #8 |
| #10 | 'Blood transfusion'/exp |
| #11 | 'transfusion' |
| #12 | #10 OR #11 |
| #13 | 'mortality'/exp |
| #14 | ((mortality AND decline OR death) AND rate OR crude) AND mortality AND rate |
| #15 | #13 OR #14 |
| #16 | 'thromboembolism'/exp |
| #17 | 'embolism'/exp |
| #18 | embolisms OR embolus |
| #19 | 'thrombosis'/exp |
| #20 | 'Venous thrombosis'/exp |
| #21 | #16 OR #17 OR #18 OR #19 OR #20 |
| #22 | 'seizure'/exp |
| #23 | #9 OR #12 OR #15 OR #21 OR #23 |
| #24 | #3 AND #6 AND #23 |

After constructing a heterogeneity matrix, the frequentist method will be applied to the fitted meta-regression model. The model includes covariates as the basic parameters and assumes that heterogeneity is independent of the comparison between effect sizes from multiarm studies. Inconsistency refers to the differences between direct and various indirect effects estimated for the same comparison. We will estimate the probability of a treatment being ranked at a specific place according to the outcome using 'network rank'.

If evidence suggests moderate statistical or clinical heterogeneity, we plan to investigate this by performing subgroup and sensitivity analyses. We will conduct subgroup and sensitivity analyses based on the actual situation of the included studies. Subgroup analysis will be performed based on sex, ethnic group, age group, sample size, type of surgery, since these factors are particularly important for dose efficacy.

**Table 4** Search strategy of Web of Science

| Search number | Query |
|---|---|
| #1 | TS= (tranexamic acid) |
| #2 | TS= (AMCHA or (t-AMCHA) or AMCA or (trans-4-(Aminomethyl) cyclohexanecarboxylic Acid) or KABI 2161 or Cyklokapron or Transamin or Spotof or Ugurol or Exacyl or Anvitoff or Amchafibrin) |
| #3 | #1 or #2 |
| #4 | TS= (Randomized Controlled Trial) |
| #5 | (TS= (Randomized Control*)) OR TS= (clinical trial*) |
| #6 | #4 or #5 |
| #7 | TS=(Hemorrhage) or AB=Bleeding or AB=Haemorrhages |
| #8 | TS= (Blood Transfusion) or AB=transfusion |
| #9 | TS=(Mortality) or AB=Mortality Decline or AB=Death Rate or AB=Crude Mortality Rate |
| #10 | TS=(Thromboembolism) or TS=Embolism or AB=Embolisms or AB=Embolus |
| #11 | TS=(Thrombosis) or TS= (Venous thrombosis) |
| #12 | TS=(Seizure) |
| #13 | #7 or #8 or #9 or #10 or #11 or #12 |
| #14 | #3 and #6 and #13 |

The Instrument for assessing the Credibility of Effect Modification Analyses tool will be used to assess the credibility of subgroup analysis.[23] Sensitivity analysis will be planned without patients undergoing cardiac surgery and without paediatric patients.

To explore the specific value of the optimal dose and better routine, the dose–response relationship will be considered in this study to determine whether there is a threshold effect.

Publication bias will be evaluated by a 'comparison-adjusted' funnel plot. GRADEpro software will be used to grade the evidence of all the outcomes, and this process will be completed by two individuals separately.

## Patient and public involvement

This study protocol did not involve either patients or the public.

## Amendments

If there are any amendments to the protocol, we will explain in the final report.

**Table 5** Search strategy of the China National Knowledge Internet

| Search number | Query |
|---|---|
| #1 | 检索范围；总库 (摘要：氨甲环酸) AND(摘要：随机对照研究) |

## ETHICS AND DISSEMINATION

No ethics approval will be sought, as no original data will be collected for this review. Findings will be disseminated through peer-reviewed publication and conference presentations.

**Contributors** XinyW: study design, conduct of study, bibliographic research, design of data entry forms, data management, protocol and manuscript writing and review. XinxW: bibliographic research design and conduct, protocol, and manuscript review. FL: protocol and manuscript review. YY: protocol and manuscript review. RH: study conception and design, scientific coordination, protocol, and manuscript writing and review.

**Funding** This study was supported by funding from the Clinical Medicine Development of Special Funding Support from the Beijing Municipal Administration of Hospitals (ZYLX201708; DFL20180502).

**Competing interests** None declared.

**Patient and public involvement** Patients and/or the public were not involved in the design, or conduct, or reporting, or dissemination plans of this research.

**Patient consent for publication** Not applicable.

**Provenance and peer review** Not commissioned; externally peer reviewed.

**ORCID iD**
Ruquan Han http://orcid.org/0000-0003-4335-8670

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
