## [Reviewer comments · BMJ Open]

ARTICLE DETAILS

TITLE (PROVISIONAL)	The safety and efficacy of intravenous or topical tranexamic acid administration in surgery: A protocol for a systematic review and network meta-analysis
AUTHORS	Wang, Xinyan; Wang, Xinxin; Liang, Fa; Yu, Yun; Han, Ruquan

VERSION 1 – REVIEW

REVIEWER	Sharif, Sameer McMaster University
REVIEW RETURNED	23-Nov-2021

GENERAL COMMENTS	Thank you so much for the opportunity to review your protocol for your systematic review and network meta-analysis on the safety and efficacy intravenous or topical tranexamic acid administration in surgery. I mostly only have minor suggestions to further improve upon your work: 1. Abstract: Minor grammatical errors. No mention of using GRADE methodology in the abstract even though it is mentioned in the manuscript. I would suggest adding that in the abstract as well.2. Introduction: Mostly well-written with minor grammatical errors. The one sentence that I believe requires re-working is the last two sentences in the introduction where you lay out the purpose of the study. 'Therefore, this study uses network meta-analysis method, relying on its characteristics of combining direct comparison and indirect comparison[20], to analyse the safety and efficacy of different doses (high, medium, low) intravenous injection or of topical application of TXA. We will try to explore the best intervention methods, reduce surgical bleeding, and improve patient outcomes.' I believe the second sentence here is expendable and can likely be removed. More importantly, however, the first sentence should be re-written so that your objective is clearer. I would suggest using the PICO format: In surgical patients of all ages, does the use of IV or topical TXA... etc... 3. Population: I see all surgical patients are being included in this. Something for you to consider: Should a sensitivity analysis be planned without Cardiac Surgery patients considering the heavy frequency with which they receive TXA?
--

	I also note that adults and pediatrics are being included and I note the pre-planned subgroup analysis based on age. This is great. Just something else for you to consider: Should there perhaps be a pre-planned sensitivity analysis without pediatric data? Furthermore, it may be valuable to evaluate subgroups based on type of surgery? 4. Protocol specifics: The protocol is pre-registered on PROSPERO which is great. Moreover, the appropriate PRISMA statement has been reviewed as well. 5. Statistics: At first glance, they do look appropriate. I appreciate random-effects being used. However, I personally use a Statistician to help with my NMAs. I have asked the BMJ to consider having a statistician review your Methods as well to ensure you are doing your best work possible. Thank you once again for allowing me to review your manuscript. I look forward to reading your completed work.
--	--

REVIEWER	Yeung, Justin University of Calgary
REVIEW RETURNED	07-Dec-2021

GENERAL COMMENTS	Congratulations on tackling a very important topic with thorough search strategy with potentially broad applications to a variety of surgical specialties. The research question is direct and specific question looking at select severe complications of TXA with succinct intervention groups. Although the article would benefit from being edited and proofread at a higher standard of English, the concepts are relatively clear and can be deduced from the current explanations. Although it is a strength to not exclude non-English papers, but accurate translation would need to be arranged and properly described in the future paper as well. I would also suggest including MESH terms 'thrombosis' or 'venous thrombosis' as part of search strategy (for example table 1 search number #7) especially since deep vein thrombosis is one of the listed outcome indicators. Or provide a rationale for not including this in the search strategy. It would be helpful to further describe the credentials of the two independent reviewers (attending, resident, student etc). The only thing I can see is that author Xinyan Wang has a Bachelor of Arts with no medical background. If there is some medical background it would help the reliability of the review. Inconsistencies are reviewed by "other authors" would be made more reliable if it was assigned to just one other author so there aren't discrepancies on how they are resolved. The intervention group is stratified by low, medium, high dose of TXA. The description of the groups can have conflicts especially when the pediatric population is not excluded. For example, a 5 kg child who uses 15mg/kg dose would result in a total infusion dose of 75 mg but would both be classified as <1g (low) and initial dose of 10-20 mg/kg (medium). How do the authors want to deal with these overlaps?
--

	Finally, in my review of the literature, the outcome measures are all over the place and not standardized which makes comparisons very difficult. Other common outcome measures to consider would include change in Hb or Hct, hematoma rates, and drain output as a surrogate of blood loss. "Intraoperative, post operative or total blood loss" is often variably defined and described. Very important topic but very big and difficult task to analyze heterogenous outcomes and studies currently available. I am personally doubtful that the outcomes available are sufficient to answer the authors primary questions. Hopefully addressing some of these issues will help streamline the study a bit more for a higher chance of success. If the authors are successful, the outcomes would likely inform or change clinical practice.
--	--

VERSION 1 – AUTHOR RESPONSE

Reviewer: 1

Dr. Sameer Sharif, McMaster University

Comments to the Author:

Thank you so much for the opportunity to review your protocol for your systematic review and network meta-analysis on the safety and efficacy intravenous or topical tranexamic acid administration in surgery. I mostly only have minor suggestions to further improve upon your work:

1. Abstract: Minor grammatical errors. No mention of using GRADE methodology in the abstract even though it is mentioned in the manuscript. I would suggest adding that in the abstract as well.

Reply: Thanks for your opinion. Grammatical errors have been corrected. GRADE methodology is added in the part of methods and analysis for *Abstract*. (The changes are marked in red).

2. Introduction: Mostly well-written with minor grammatical errors. The one sentence that I believe requires re-working is the last two sentences in the introduction where you lay out the purpose of the study.

'Therefore, this study uses network meta-analysis method, relying on its characteristics of combining direct comparison and indirect comparison [20], to analyze the safety and efficacy of different doses (high, medium, low) intravenous injection or of topical application of TXA. We will try to explore the best intervention methods, reduce surgical bleeding, and improve patient outcomes.'

I believe the second sentence here is expendable and can likely be removed. More importantly, however, the first sentence should be re-written so that your objective is clearer. I would suggest using the PICO format:

In surgical patients of all ages, does the use of IV or topical TXA... etc...

Reply: Appreciated for your opinion. The last sentence in *Background* has been rewritten using the PICO format. (The changes are marked in red).

3. Population: I see all surgical patients are being included in this. Something for you to consider: Should a sensitivity analysis be planned without *Cardiac Surgery patients* considering the heavy frequency with which they receive TXA?

I also note that *adults and pediatrics* are being included and I note the pre-planned subgroup analysis based on age. This is great. Just something else for you to consider: Should there perhaps be a pre-planned sensitivity analysis without pediatric data?

Furthermore, it may be valuable to evaluate subgroups *based on type of surgery*?

Reply: Thanks for your opinion. Pre-planned subgroup for type of surgery and sensitivity analysis without pediatric data or cardiac Surgery patients have been added into the third paragraph of *Statistical analysis*. (The changes are marked in red).

4. Protocol specifics: The protocol is pre-registered on PROSPERO which is great. Moreover, the appropriate PRISMA statement has been reviewed as well.

Reply: Thanks for your comments. We have updated the file of *PRISMA statement* based on the newest manuscript.

6. Statistics: At first glance, they do look appropriate. I appreciate random-effects being used. However, I personally use a Statistician to help with my NMAs. I have asked the BMJ to consider having a statistician review your Methods as well to ensure you are doing your best work possible.

Reply: Thanks for your comments. Your suggestion is extremely helpful.

Reviewer: 2

Dr. Justin Yeung, University of Calgary

Q1: Although the article would benefit from being edited and proofread at a higher standard of English, the concepts are relatively clear and can be deduced from the current explanations. Although it is a strength to not exclude non-English papers, but accurate translation would need to be arranged and properly described in the future paper as well.

Reply: Thanks for your comments. This is a particularly good suggestion. We will translate properly. If we encounter non-English and non- Chinese articles in the included studies, we will seek language help from professionals.

Q2: I would also suggest including MESH terms 'thrombosis' or 'venous thrombosis' as part of search strategy (for example table 1 search number #7) especially since deep vein thrombosis is one of the listed outcome indicators. Or provide a rationale for not including this in the search strategy.

Reply: Thanks for your opinion. MESH terms 'thrombosis' or 'venous thrombosis' have been added into the search strategy (in table1,2,3,4). The changes are marked in red.

Q3: It would be helpful to further describe the credentials of the two independent reviewers (attending, resident, student etc.). The only thing I can see is that author Xinyan Wang has a Bachelor of Arts with no medical background. If there is some medical background it would help the reliability of the review. Inconsistencies are reviewed by "other authors" would be made more reliable if it was assigned to just one other author so there aren't discrepancies on how they are resolved.

Reply: Thanks for your suggestion. When our first author prepared this article, she was about to graduate from a medical bachelor's degree, so she wrote it as a bachelor's degree. We apologize for the misinterpretation of B.A. 's description. Now, the first author has nearly entered the stage of postgraduate study, and we think M.D. may be more appropriate. We have corrected the author information. Thank you for reminding us.

Q4: The intervention group is stratified by low, medium, high dose of TXA. The description of the groups can have conflicts especially when the pediatric population is not excluded. For example, a 5 kg child who uses 15mg/kg dose would result in a total infusion dose of 75 mg but would both be classified as <1g (low) and initial dose of 10-20 mg/kg (medium). How do the authors want to deal with these overlaps?

Reply: In intravenous administration, the usual dosing method is based on mg/kg, especially for children, where accurate dosing is more important. When such conflicts arise (just like your example), the study is included in the low-dose group. Sensitivity analysis without pediatric data or has been added into the third paragraph of *Statistical analysis*.

Q4: Finally, in my review of the literature, the outcome measures are all over the place and not standardized which makes comparisons very difficult. Other common outcome measures to consider would include change in Hb or Hct, hematoma rates, and drain output as a surrogate of blood loss. "Intraoperative, post operative or total blood loss" is often variably defined and described. Reply: Thanks for your suggestions. This is a very big challenge. We have added the outcome measures including change in Hb or Hct, hematoma rates, and drain output.

Very important topic but very big and difficult task to analyze heterogenous outcomes and studies currently available. I am personally doubtful that the outcomes available are sufficient to answer the authors primary questions. Hopefully addressing some of these issues will help streamline the study a bit more for a higher chance of success. If the authors are successful, the outcomes would likely inform or change clinical practice.

Reviewer: 1
Competing interests of Reviewer: None

Reviewer: 2
Competing interests of Reviewer: None

VERSION 2 – REVIEW

REVIEWER	Sharif, Sameer McMaster University
REVIEW RETURNED	22-Feb-2022

GENERAL COMMENTS	Thank you for taking my input into account with your updated manuscript. I just have minor suggestions once more as I do not think one of my more important points of question were fully addressed. 1. Abstract - GRADE sentence: - Change to "We will assess the overall certainty of evidence for each outcome using the Grading Recommendations Assessment, Development and Evaluation approach." - Please make this change to Point 4 of the Strengths and Limitations of the study as well 2. Introduction - Last sentence, Consider re-writing as follows: - In this network meta-analysis, we plan on analyzing the safety and efficacy of different doses (high, medium, low) of intravenous injection or topical application of TXA in surgical patients of all ages, using direct and indirect comparisons [20]. 3. Outcomes - Sorry I did not catch this earlier but it would be preferable if there were around 6-7 outcomes being recorded as per GRADE methodology - It would be ideal if you could streamline some of your outcomes 4. Subgroups - To improve credibility of subgroups, you could use the ICEMAN tool and you could also hypothesize which direct the treatment effect will be - See this link: https://pubmed.ncbi.nlm.nih.gov/32778601/
---

	5. Grammar  - I think prior to re-submission/publication, it would be prudent to have this proofread once more to ensure there are no more grammatical errors (as there are several significant errors) - This was brought up by the other reviewer but was not addressed - I would strongly suggest getting this proofread at a higher standard of English and then re-submitting Thank you.
--	--

VERSION 2 – AUTHOR RESPONSE

Reviewer Report:

Reviewer: 1

Dr. Sameer Sharif, McMaster University

Comments to the Author:

Thank you for taking my input into account with your updated manuscript. I just have minor suggestions once more as I do not think one of my more important points of question were fully addressed.

1. Abstract - GRADE sentence:

- Change to "We will assess the overall certainty of evidence for each outcome using the Grading Recommendations Assessment, Development and Evaluation approach."
- Please make this change to Point 4 of the Strengths and Limitations of the study as well

Reply: Thanks for your comments. Thank you for helping us improve the language description. (The changes are marked in red)

2. Introduction - Last sentence, Consider re-writing as follows:

- In this network meta-analysis, we plan on analyzing the safety and efficacy of different doses (high, medium, low) of intravenous injection or topical application of TXA in surgical patients of all ages, using direct and indirect comparisons [20].

Reply: Thanks for your comments. We have rewritten the last sentence according to your suggestion, a more concise version. (The changes are marked in red)

3. Outcomes

- Sorry I did not catch this earlier but it would be preferable if there were around 6-7 outcomes being recorded as per GRADE methodology

- It would be ideal if you could streamline some of your outcomes

Reply: Thanks for your comments. It is a good idea to have a limited number of outcomes; however, each study has different indicators for expressing bleeding for efficacy outcome. We have choose some important outcomes in the for GRADE assessment. (The changes are marked in red in the methods section)

1)Efficacy outcomes: bleeding volume, blood transfusion volume, change in Hb or Hct.

2) Safety outcomes: mortality, incidence of thromboembolism, seizures.

4. Subgroups

- To improve credibility of subgroups, you could use the ICEMAN tool and you could also hypothesize which direct the treatment effect will be.

- See this link: <https://pubmed.ncbi.nlm.nih.gov/32778601/>

Reply: Thanks for your comments. It is a good idea to have a useful tool to assess the subgroup analyses. We have added the description in the section on 'statistical analysis and the references in the list. (The changes are marked in red)

5. Grammar

- I think prior to re-submission/publication, it would be prudent to have this proofread once more to ensure there are no more grammatical errors (as there are several significant errors)

- This was brought up by the other reviewer but was not addressed

- I would strongly suggest getting this proofread at a higher standard of English and then re-submitting

Reply: Thanks for your comments. This resubmitted version has been reviewed by professional language editors. We have uploaded the editing certification in the supplemental material.

Thank you.

Reviewer: 1

Competing interests of Reviewer: None